# Sexual and reproductive health outcomes of women who experienced violence in Germany: Analysis of the German health interview and examination survey for adults (DEGS1)

**Antonia Marie Wellmann[1,2], Diogo Costa [1,3]***

1 Department of Population Medicine and Health Services Research, School of Public Health, Bielefeld University, Bielefeld, Germany, 2 Deutsche Gesellschaft für Internationale Zusammenarbeit (GIZ), Eschborn, Germany, 3 Research Centre for Human Development (CEDH), Faculty of Education and Psychology, Universidade Católica Portuguesa, Porto, Portugal

* diogo.costa@uni-bielefeld.de, dancosta@ucp.pt

**Data Availability Statement:** "The data that support the findings of this study are owned by a third-party organization. The data are available from the Robert Koch Institute (available as

## Abstract

### Objectives

Violence against women is a widespread public health concern with severe effects to women's sexual and reproductive health, including higher risks for miscarriage or stillbirth, unintended pregnancy and induced abortion. This study examined the association between women exposure to physical violence, psychological violence and sexual and reproductive health outcomes (contraceptive use, miscarriage or stillbirth and abortion) in Germany.

### Methods

This study used a cross-sectional research design to analyze data on violence against women and sexual and reproductive health (SRH) outcomes collected through the German Health Interview and Examination Survey for Adults, Wave 1, between 2008 and 2011 (n = 3149 women, aged 18–64 years). Multivariable logistic regression models were used to assess the association between experiences of violence among women and the presence of sexual and reproductive health outcomes, considering the influence of socio-demographic and health-related factors (age, marital status, socioeconomic status, social support, number of children, alcohol consumption, health status, chronic conditions).

### Results

Three associations remained significant (p<0.05) in fully-adjusted models: (i) exposure to physical violence by a parent or caregiver and birth control pill utilization (aOR, adjusted Odds Ratio, 95% CI: 1.36, 1.02–1.81) (ii) exposure to physical violence since the age of 16 and miscarriage or stillbirth (aOR, 95%CI: 1.89, 1.17–3.04); and (iii) exposure to psychological violence by a parent or caregiver and abortion (aOR, 95%CI: 1.87, 1.30–2.70).

Scientific Use File) but restrictions apply to the availability of these data, which were used under license for the current study, and so are not publicly available. Further information including instructions on how to request a Scientific Use File of these data can be found in the RKI website (https://www.rki.de/DE/Content/Forsch/FDZ/Zugang/SUF.html)."

**Funding:** The author(s) received no specific funding for this work.

**Competing interests:** The authors have declared that no competing interests exist.

## Conclusions

The results suggest that adult German women who experienced physical or psychological violence since the age of 16, including violence perpetrated by a parent or caregiver, were more likely to report miscarriage or stillbirth and abortion. Direct assessment of violence experiences against women should be conducted by healthcare professionals in clinical encounters, particularly by obstetrics and gynaecological specialists, for the prevention of women´s adverse sexual and reproductive health outcomes. Furthermore, violence should be treated as a major public health concern and addressed through a multisectoral approach, involving the healthcare and educational sectors, researchers and relevant policymakers.

## Introduction

Violence against women is common all over the world [1, 2], and recognized as a global public health problem and a human rights violation [1, 3–5]. Violence against women includes acts of physical, psychological and sexual violence but also harassment, stalking, violence during childhood and harmful traditional practices, such as female genital mutilation (FGM) [2, 6–8]. Intimate partner violence (IPV) is the most common form of violence against women, encompassing acts of physical, psychological and/or sexual violence perpetrated by a current or former partner [2]. Around one in three women and girls (35%) worldwide have experienced either physical and/or sexual IPV or non-partner sexual violence at some point in their life [2]. In Europe, around 27% of women have reported experiences of IPV and/or non-partner sexual violence in their life [2], and about 8% have experienced physical and/or sexual violence in the past 12-months [9]. For childhood abuse (commonly defined as the experience of acts of physical and/or sexual abuse before the age of 15), some 27% of women in Europe have experienced some form of physical violence and 12% has experienced some form of sexual abuse by an adult [9].

In Germany, 35% of women have experienced violence (by any partner and/or non-partner) since the age of 16 and 44% experienced some form of violence (physical, sexual or psychological abuse) during childhood [9]. Despite various research efforts, documenting the real magnitude of violence experienced by women is hindered by a lack of reliable and comparable data. Violent incidents mainly happen in private settings and many women do not report to authorities, seek help or talk openly about their experiences [1, 10].

Violence has serious short-, medium-, and long-term consequences to the physical and mental health and wellbeing, health behaviors, and health opportunities of women [11]. Health consequences include direct harm, such as physical injuries, but also indirect consequences, such as psychological trauma and stress which makes violence an important determinant of women's morbidity and mortality [2, 9, 12].

Adverse sexual and reproductive health (SRH) outcomes can result from violence due to psychological stress, risk behavior, physical injuries and limited access to healthcare. These include higher risks for miscarriage or stillbirth, as well as unintended (adolescent) pregnancy, resulting in a higher risk for induced abortion [13–17]. According to the WHO, women who experienced violence are more than twice as likely to have an abortion than women who never experienced violence [2, 10]. Pallitto and colleagues (2013) [18] estimated that women with a history of IPV have a 1.5-fold increased risk for stillbirth as well as a 2-fold increased risk in pregnancy loss compared to women who have no history of IPV.

In a longitudinal analysis conducted in Australia during 2021–2022 among two cohorts, IPV exposure was shown associated with increased odds of experiencing several sexual and reproductive health outcomes, including infertility, termination and miscarriage, which showed a dose-response effect, i.e., increased odds of experience adverse outcomes with greater exposure to IPV [19].

A recent systematic review on IPV and perinatal health included 15 studies conducted in Asia, 12 in North America and Oceania, 12 in Central and South America, 6 in Africa and 5 in Europe, to study the relationship between IPV and 39 different outcomes [20]. Amongst the studies classified with high methodological quality, ten studies included miscarriage as an outcome and six of them found an association with IPV (with miscarriage risks ranging between 1.7 and 5.7 times higher among women experiencing IPV). In the same review, among seven studies with high methodological quality that reported outcomes of perinatal death, six studies associated IPV experiences with this outcome (with risk ranging between 1.4 and 6.04 times higher) [20].

Regarding the association between violence and women's contraceptive use, several studies have shown a reduction in use of modern contraception (includes birth control pills, IUD, injection, condom, male or female sterilization) and an increased risk for unmet need for family planning among women who experienced past-year or lifetime violence [21–24]. A systematic review and meta-analysis that included longitudinal, RCT, and case-control studies examining the association between IPV and women's contraceptive use, suggests that IPV is associated with a decrease in women's use of contraceptive methods. Other studies (with cross-sectional designs), show that women ever exposed to violence are more likely to use contraceptives than women who were never exposed to violence [25–28], possibly explained by a greater will to avoid pregnancy [29, 30]. Further studies are needed to better elucidate the associations between different forms of violence experienced by women and contraceptive use.

In Germany, a recent exploration of the German Health Interview and Examination Survey for Adults (DEGS1) ("Studie zur Gesundheit Erwachsener in Deutschland") [31], unraveled that victims of violence visited their general practitioner more often, compared to non-victims, but this study did not explore the relationship between women who experienced violence and SRH outcomes, considering different violence forms (e.g. physical, psychological) and periods of exposure (e.g. past-year, lifetime, or during childhood). Hence, the main goal of this work was to assess the association of different forms of violence experienced by women with different SRH, based on representative population-based data from the German Health Interview and Examination Survey for Adults (DEGS1) ("Studie zur Gesundheit Erwachsener in Deutschland") [32] which was conducted between 2008 and 2011 by the Robert Koch-Institute (RKI). These contextualized results can help raise awareness about the adverse effects that violence experiences have in SRH outcomes, and about the need to address this global public health issue through a multisectoral approach that aims at violence prevention.

## Methods

The data analyzed in this work was gathered in the scope of the "German health interview and examination survey for adults" (DEGS), which is part of the health monitoring program of the Robert Koch Institute (RKI). The first wave of the survey (DEGS1) was conducted from 2008 to 2011. The goal of the survey was to gather representative data to describe population health and estimate disease prevalence of adults in Germany and thus contribute to evidence-based decisions in the healthcare system [32–34].

## Survey population and participant recruitment

The survey population of DEGS1 were adults aged 18 to 79 years (n = 8,152) having their main residence in Germany, including German citizens, people of foreign nationality and people living in institutions such as residential homes [34, 35]. In order to achieve a representative sample of the population, a two-stage stratified (lumped) sample was drawn as detailed elsewhere (original study in German, also available as an English version in S1 File) [35]. In the first stage, 180 study locations were selected from a total set of communities in Germany, taking federal state and community type into consideration [34–36]. Stratification weights ensured that each community was drawn with a probability proportional to its number of inhabitants. In the second stage, 11,008 people stratified by age were randomly drawn from the address registers of the Resident Registration Office. Additionally, 7,124 former participants from the 1998 Federal Health Survey (BGS98)–a previous nationwide cross-sectional study of adult health conducted by the RKI–were contacted to allow not only cross-sectional but also longitudinal analysis [33, 34].

Overall, 8,152 men and women participated in the survey. Of these, 4,193 were first-time participants and 3,959 were people who had already participated in BGS98. A total of 17,410 men and women between the ages of 18 and 91 were invited, resulting in response rates of 42% for the newly recruited participants and 62% for the former participants of BGS98 [33, 35, 37].

## Ethical considerations

The DEGS1 study protocol was approved by the Medical Ethics Review Committee of the Charité in Berlin—No. EA2/047/08 [37].

The screening instrument for violence was successfully tested in a DEGS1 pretest and clinically validated on a sample of 830 patients at the outpatient clinic of the Department of Psychosomatic Medicine and Psychotherapy at the University of Düsseldorf [33]. The collection of data regarding sensitive topics, including violence victimization and its outcomes, requires the consideration of additional ethical and safety aspects [33]. This includes the anonymity and confidentiality of all study participants as well as the training and support of staff as they might spend a significant time with study participants and learn many personal details. Also, the potential re-traumatization of participants triggered by the survey has to be taken into consideration [38]. Participants were therefore questioned with the help of a self-completion questionnaire which they filled out at an out-of-home examination center as part of a wider health examination survey, to ensure the safety of participants. The questionnaires were not identified by name. In case participants experienced mental stress resulting from the survey, contact information of help services were made available. Due to the possibly high stress for the participants and a lack of possibilities for crisis intervention in the examination setting, victimization of sexual violence was not included in the survey [32, 33].

Participants provided written informed consent prior to the interview and examination. The survey was conducted in health examination centers and questions addressing women´s experiences of violence were self-administered. Staff was trained and supervised. Local crisis lines and emergency addresses were available in case of potential re-traumatization.

## Survey instruments and data handling

The survey obtained cross-sectional data on violence victimization in adulthood and childhood by a self-completion pen-and-paper questionnaire in the age range of 18 to 64 years among a total of 5,939 participants, of whom 3,149 were women. The questionnaire for the screening of physical and psychological violence was developed by the RKI in cooperation with researchers from Bielefeld University [33].

The questionnaire explicitly did not mention the word violence. Questions about this topic were introduced as follows: "Sometimes people are attacked, get into physical fights or have experiences that they find hurtful or upsetting. The following questions are about these specific experiences in your everyday life." The questions concerning violence victimization included physical and psychological violence in the past 12 months, since the age of 16, as well as physical and psychological violence experienced during childhood and adolescence before the age of 16 by a parent or caregiver and by peers (here defined as childhood violence, questions shown in S1 Table in S1 File). The 12-month prevalence and the prevalence since the age of 16 were measured with a binary (yes or no) answering option. Childhood violence experiences variables were measured on a 4-option Likert scale (Frequently, Occasionally, Rarely, Never), and categorized into binary variables as follows: yes = for options "Frequently, Occasionally, Rarely"; and no = for option "Never".

As SRH outcomes, lifetime usage of birth control pill, current contraceptive use, lifetime miscarriage or stillbirth, and lifetime abortion were considered and assessed with the questions "Have you ever taken the birth control pill? (yes/no)", "Are you currently using contraceptives? (yes/no)", and "Please tell us how many live births, miscarriages, stillbirths, and abortions you have had", as detailed in S1 Table in S1 File).

The age-group categories used for analysis were: 18 to 24 years, 25 to 34 years, 35 to 44 years, 45 to 54 years and 55 to 64 years. For family status the answering options (also corresponding to the four categories analyzed) were: "Married, living with spouse; Married, living separated from spouse; Single; Divorced or Widowed".

Self-rated health status was assessed using the first question from version 2.0 of the SF-36 Health Survey [32], including a five-tier Likert-Scale answering option, "Very good; Good; Average, Poor or Very poor", that was categorized for analysis in two as "Very good or good" and "Average, Poor or Very poor".

To assess alcohol consumption, three questions of the Alcohol Use Disorders Identification Test-Consumption (AUDIT C) were asked [32], including questions on how often a participant drinks alcohol and how much they drink in a day or on an occasion ("How often did you have a drink containing alcohol in the past year?", answering options: "Never", "Monthly or less", "Two to four times a month", "Two to three times a week", and "Four or more times a week"; "How many drinks did you have on a typical day when you were drinking in the past year?", answering options: "None, I don't drink", "1 or 2", "3 or 4", "5 or 6", "7 or 9", "10 or more"; "How often did you have six or more drinks on one occasion in the past year?", answering options: "Never", "Less than monthly", "Monthly", "Weekly", and "Daily or almost daily"). Each question is scored from 0 to 4, and the scores are summed for a possible score of 0 to 12. Scores of "0" are categorized "never" drinkers. Category "risk consumption" is defined as $> 3$ in women and $> 4$ in men, and the remaining values are categorized as "moderate".

For social support the Oslo-3 Item Social Support Scale was used [32], asking, a) how many people the participants can rely on for help with serious private problems (answering options: "none", "1–2", "3–5", "5+"), b) how much interest and sympathy other people show in what they do ("none", "little", "uncertain", "some", "a lot"), and c) how easy it is to get practical help from neighbors ("very difficult", "difficult", "possible", "easy", "very easy"). The total sum score ranges from 3 to 14, and was categorized into "low" support (scores from 3 to 8), "medium" support (scores from 9 to 11), and "high" support (scores from 12 to 14).

Socioeconomic status (SES) was assessed with a previously developed SES Index encompassing information on participant education, income and occupation [39] and categorized in a three-tier scale (low, middle, and high SES).

Regarding chronic conditions the question was "Do you have one or more long-lasting, chronic diseases? Chronic diseases are long-lasting conditions that require constant treatment

and monitoring, such as diabetes or heart disease'" with the options to answer "Yes", "No", "Don't know" (the latter two options were merged for analysis in the same category).

Participants were also asked if and how often they have consulted physicians in private practices or home visits in the past 12 months, with gynecologist being one of the answering options. As part of a physical examination, participants were also screened for chronic conditions, including diabetes and hypertension [32], further categorized in two binary variables signaling the presence/absence of these diagnosis.

Participant's number of children was asked and categorized in 1, 2, 3 or more.

## Statistical analysis

Analysis was restricted to women under 65 years old, since the violence module and the SRH outcomes of interest were only administered to this group. Analysis was conducted using the IBM SPSS Statistics software version 27.0.0.0. For all statistical analyses, a p-value of $<0.05$ was considered as significant.

DEGS1 aims to be representative of the national population of adults. In order to obtain estimates that are representative for the German population, population weights must be considered in all cross-sectional statements on prevalence and distributions [36]. These weights correct for variations of the sample from the population structure with respect to various characteristics, including age, gender or level of education [35].

Due to the complex sampling design, sample points and participants had unequal probabilities of selection. Design weights were developed to compensate for this selection probability as well as nonresponse biases. Furthermore, adjustment weights were calculated by comparing the collected data with data from a non-participant short survey as well as data from representative demographic statistics of the German population by the Federal Statistical Office [35, 36]. A cluster variable considers that the subjects were drawn within sample points and that participants within a community are likely to be more similar than participants from different communities. The population weight and the cluster variable were included in all the analysis by using the complex samples function of SPSS.

We calculated absolute and relative frequencies of violence victimization, SRH outcomes as well as key demographics and characteristics of the participants to assess the overall composition of the sample. The relative frequencies, moreover, indicated the prevalence of the various forms of violence and the SRH outcomes.

Weighted $\chi^2$ tests were calculated to evaluate the association between the socio-demographic and health-related characteristics and the predictor variables (12-month physical and psychological violence, lifetime physical and psychological violence, physical and psychological childhood violence of parents and peers) as well as the association between the socio-demographic, health-related characteristics and the SRH outcomes (lifetime usage of birth control pill, current contraceptive use, lifetime miscarriage or stillbirth, lifetime abortion), and are presented as S1 File. $\chi^2$ tests were also calculated to assess the association between the violence variables and the SRH outcomes.

Multivariable logistic regression models were used to measure the magnitude of associations between experiences of violence among women (compared to not having experienced violence) and the presence of SRH outcomes while taking the influence of confounders into account. Weighted Odds Ratios with respective 95% confidence intervals (CI) were used to estimate the magnitude of associations between predictors and outcomes.

The effect of the confounders was tested in a stepwise approach. Possible confounders were grouped based on literature that previously identified such factors as determinants of violence frequency and/or SRH outcomes, and according to their thematic category (e.g. age, marital

status and socioeconomic status were categorized as "demographics of women"). Five groups were thus considered and entered in five sequential models (multivariable models means that each model has one outcome variable and multiple predictors):

Model 1: demographics (age; marital status; socioeconomic status);

Model 2: social structure (social support; number of children);

Model 3: health-status and behaviors (health-status, alcohol consumption, chronic condition);

Model 4: utilization of prevention services (Visited a gynecologist in the last 12 months; cervix cancer smear (ever));

Models 5: additional chronic conditions (diabetes, hypertension).

The groups of confounders which were found to influence the association between the violence variables and the SRH variables, e.g. by changing the level of significance or the magnitude of the odds ratio, were kept in the model. Variables not changing the magnitude of the association or significance were not retained. Based on this process, fully adjusted models included age, marital status, social support, health status, number of children, socioeconomic status (SES), alcohol consumption and chronic conditions as confounders (i.e., Model 3). Results from models further adjusted for utilization of prevention services (visited a gynecologist in the last 12 months, cervix cancer smear (ever) and additional chronic conditions (diabetes and hypertension) (i.e., Models 4 and 5) are presented as S1 File–these were not retained as no evidence of their influence on exposure or outcome was found, and due to collinearity with "chronic conditions" in the case of diabetes and hypertension.

In total, we present 32 final adjusted models, correspondent to the stratification according to 8 types of violence (physical violence in the past 12 months, psychological violence in the past 12 months, physical violence since the age of 16, psychological violence since the age of 16, physical violence by a parent/caregiver, psychological violence by a parent/caregiver, physical violence by peers, psychological violence by peers), and according to 4 SRH outcomes (lifetime usage of birth control pill, current contraceptive use, lifetime miscarriage or stillbirth, and lifetime abortion).

## Results

### Sample characteristics

Overall, 3,149 women under the age of 65 answered the questionnaire on violence victimization. 48.5% (weighted %) of women were 45 years or older, almost two thirds (58.9%, weighted %) of women were married and living with their spouse and most women were classified as having a medium socioeconomic status (SES) (62.3%), Table 1. Furthermore, 78.1% of women described their health status as good or very good and about half of the participants scored a medium social support scale (49.0%) and a moderate alcohol consumption classification (57.2%). 32.3% of women had no children and 21.8% had one child. 25.0% of women reported a chronic condition.

Furthermore, about 20.2% (weighted %) of women experienced psychological and 3.3% experienced physical violence in the 12 months prior to the survey. 9.6% and 22.2% of women reported at least one experience that could be classified as physical and psychological violence since the age of 16. During childhood 36.0% reported an incident of physical violence by a parent or caregiver and 31.6% by a peer. Psychological violence by a parent or caregiver were experienced by 25.9% and psychological violence by a peer were experienced by 30.0% of women, Table 2 and Fig 1.

Regarding the SRH outcomes, 83.3% (weighted %) of participants ever took the birth control pill and 43.6% still used contraceptives of any kind at the point of the survey. About 18.5%

**Table 1. Socio-demographic characteristics of the sampled women.**

| Socio-demographic characteristics | | % weighted* (95%CI) |
|---|---|---|
| **Age** | 18–24 years | 12.9 (11.6–14.3) |
| | 25–34 years | 18.9 (17.1–20.9) |
| | 35–44 years | 22.3 (20.6–24.2) |
| | 45–54 years | 25.7 (23.9–27.6) |
| | 55–64 years | 20.1 (18.7–21.6) |
| **Marital Status** | married | 58.9 (56.4–61.4) |
| | married but separated | 2.4 (1.7–3.2) |
| | single | 27.9 (25.9–30.00) |
| | divorced or widowed | 10.8 (9.5–12.3) |
| **Socioeconomic Status** | low | 17.8 (16.1–19.7) |
| | medium | 62.3 (60.0–64.5) |
| | high | 19.9 (18.0–21.9) |
| **Overall health status** | very good/good | 78.1 (76.2–79.9) |
| | medium/bad/very bad | 21.9 (20.1–23.8) |
| **Social Support** | low | 10.1 (8.8–11.6) |
| | medium | 49.0 (46.8–51.2) |
| | high | 40.9 (38.8–43.0) |
| **Alcohol consumption** | never | 15.3 (13.7–17.1) |
| | moderate | 57.2 (54.8–59.6) |
| | risk consumption | 27.5 (25.4–29.6) |
| **Number of children** | 0 | 32.3 (30.3–34.3) |
| | 1 | 21.8 (20.2–23.6) |
| | 2 | 32.1 (30.2–34.0) |
| | 3+ | 13.8 (12.3–15–4) |
| **Chronic conditions** | any | 25.0 (23.1–26.9) |
| | none | 75.0 (73.1–76.9) |

* Population weights according to [36]; 95%CI = 95% Confidence Interval

of the surveyed women ever had a miscarriage or stillbirth and 17.0% ever had an abortion, Table 3 and Fig 2.

Results from the bivariate analyses conducted for the experiences of violence and sociodemographic factors, and between each SRH and sociodemographic factors are presented as S1 File.

## Experiences of violence and sexual and reproductive health outcomes

Women who reported to have experienced physical violence in the prior 12 months answered significantly more often (p>0.05) to ever have taken the birth control pill (weighted %: 70.6%, 95%CI = 56.9–81.4%) than women who reported to not have experienced physical violence in the past 12 months (83.7%, 95%CI = 81.9–85.3%), Table 4. Moreover, women who indicated to have experienced at least one incident of physical violence by a parent or caregiver during their childhood were significantly more likely (p>0.05) to ever have used the birth control pill compared to women who answered to never have experienced physical violence by a parent or caregiver during childhood. For the other types of violence included in the survey, no statistically significant differences were found according to the use of birth control pill.

**Table 2. Experiences of violence in the sample.**

| Violence experiences | | %, weighted* (95%CI) |
|---|---|---|
| **Physical violence 12 months** | no | 96.7 (95.8–97.4) |
| | yes | 3.3 (2.6–4.2) |
| **Psychological violence 12 months** | no | 79.8 (78.0–81.5) |
| | yes | 20.2 (18.5–22.0) |
| **Physical violence since the age of 16** | no | 90.4 (89.0–91.7) |
| | yes | 9.6 (8.3–11.0) |
| **Psychological violence since the age of 16** | no | 77.8 (75.8–79.6) |
| | yes | 22.2 (20.4–24.2) |
| **Physical violence by parent/caregiver** | never | 64.0 (61.8–66.2) |
| | ever | 36.0 (33.8–38.2) |
| **Psychological violence by parents/caregiver** | never | 74.1 (72.0–76.1) |
| | ever | 25.9 (23.9–28.0) |
| **Physical violence by peers** | never | 68.4 (66.2–70.5) |
| | ever | 31.6 (29.5–33.8) |
| **Psychological violence by peers** | never | 70.0 (67.9–72.1) |
| | ever | 30.0 (27.9–32.1) |

* Population weights according to [36]; 95%CI = 95% Confidence Interval; Age and socioeconomic status comparison between participants with full valid information and those with at least one missing value in one of the violence variables considered is presented in S1 File.

Current contraceptive use was significantly higher (p>0.05) among women who declared to have been exposed to psychological violence during the past 12 months (50.4%, 95% CI = 45.1–55.6%) compared to women who declared to not have been exposed to

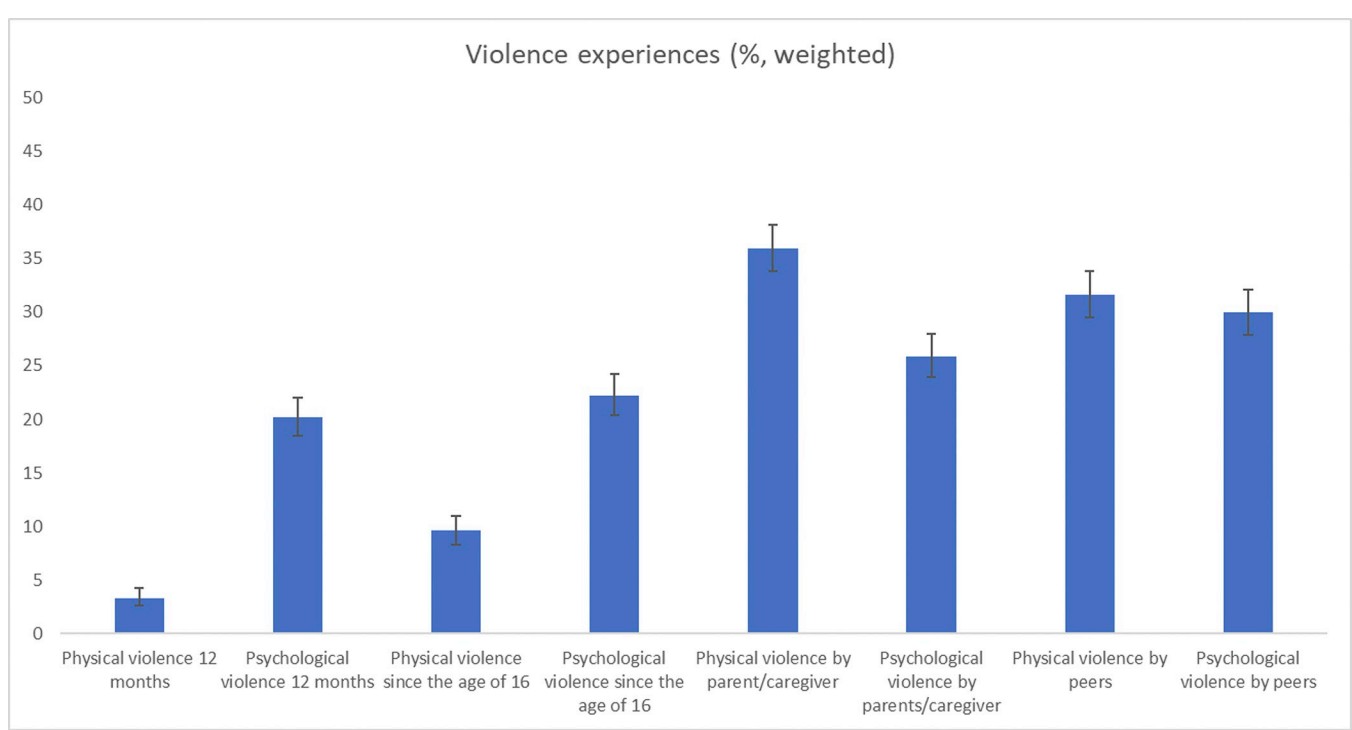

**Fig 1. Violence experiences (%, weighted).**

**Table 3. Sexual and reproductive health outcomes of the surveyed women.**

| Sexual and reproductive health | | %, weighted* (95%CI) |
|---|---|---|
| **Birth control pill (ever)** | no | 16.7 (15.1–18.4) |
| | yes | 83.3 (81.6–84.9) |
| **Current contraceptive use** | no | 56.4 (54.4–58.4) |
| | yes | 43.6 (41.6–45.6) |
| **Ever miscarriage or stillbirth** | no | 81.5 (79.7–83.1) |
| | yes | 18.5 (16.9–20.3) |
| **Ever abortion** | no | 83.0 (80.8–84.9) |
| | yes | 17.0 (15.1–19.2) |

* Population weights according to [36]; 95%CI = 95% Confidence Interval

psychological violence in the past 12 months (41.9%, 95%CI = 39.7–44.2%). Furthermore, 48.6% (95%CI = 44.3–52.9%) of women who stated to have been exposed to some form of psychological violence since the age of 16, stated to use some method of contraception at the time of the survey. This was a significantly greater proportion compared to those who answered to not have experienced any psychological violence since the age of 16 (42.5%, 95%CI = 40.1–45.0%). A significant difference (p>0.001) regarding current contraceptive use was also found between women who reported to ever have experienced physical violence by a parent or caregiver during their childhood (38.2%, 95%CI = 34.4–42.1) and women who reported to never have experienced any physical violence by a parent or caregiver (46.6%, 95%CI = 44.0–49.2%). Additionally, the results showed a statistically significant difference for the frequency of

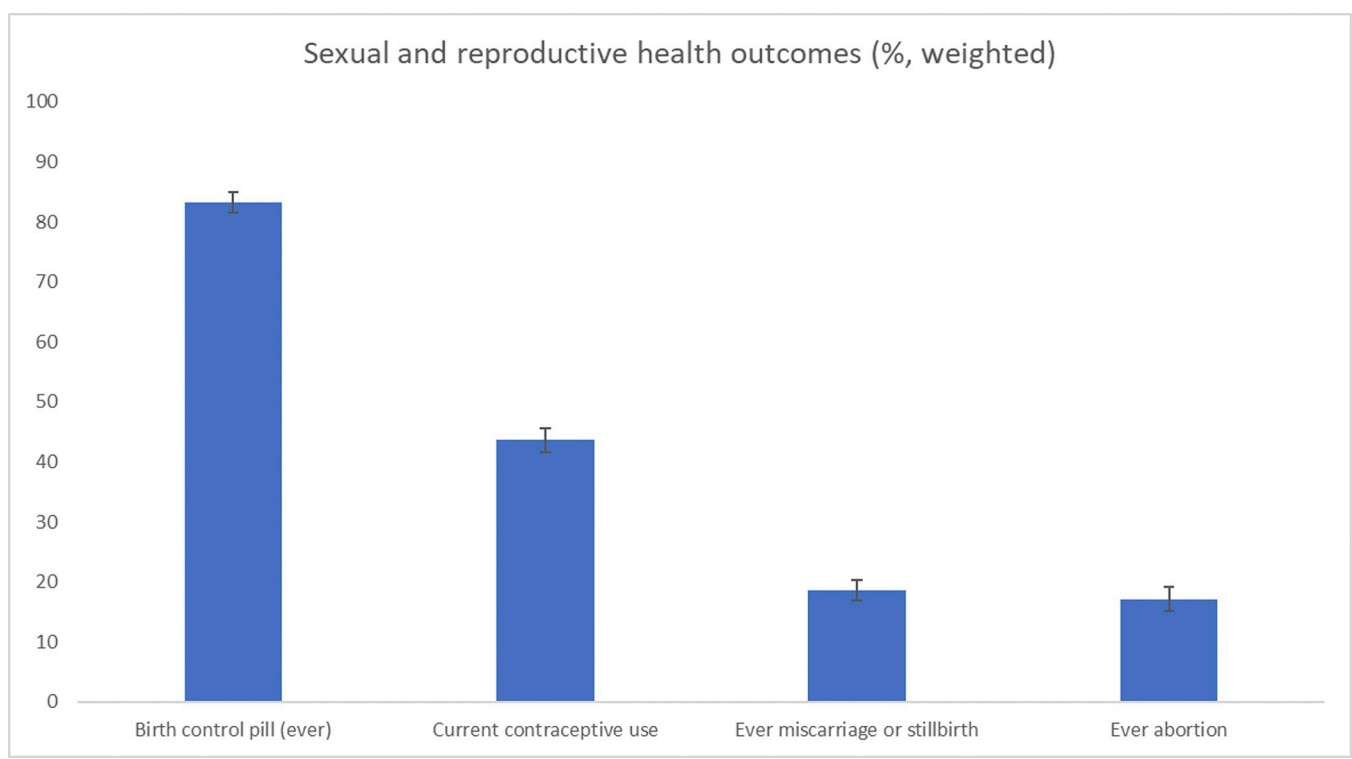

**Fig 2. Sexual and reproductive health outcomes (%, weighted).**

**Table 4. Sexual and reproductive health outcomes according to type of violence.**

| Violence | Sexual and reproductive health outcome | | | | | | | | | | | |
|---|---|---|---|---|---|---|---|---|---|---|---|---|
| | Birth control pill (ever) | | | Current contraceptive use | | | Ever miscarriage or stillbirth | | | Ever abortion | | |
| | n, yes | Weighted* % (95%- CI) | Weighted* χ² | n, yes | Weighted* % (95% CI) | Weighted* χ² | n, yes | Weighted* % (95% CI) | Weighted* χ² | n, yes | Weighted* % (95%- CI) | Weighted* χ² |
| **Physical violence 12 months** | | | **.015** | | | .213 | | | .390 | | | .857 |
| no | 2,554 | 83.7 (81.9–85.3) | | 1161 | 43.4 (41.4–45.4) | | 466 | 18.7 (17.0–20.5) | | 455 | 17.0 (15.0–19.2) | |
| yes | 59 | 70.6 (56.9–81.4) | | 39 | 51.3 (38.8–63.7) | | 10 | 13.9 (6.8–26.4) | | 9 | 16.0 (8.0–19.5) | |
| **Psychological violence 12 months** | | | .966 | | | **.005** | | | **.002** | | | .266 |
| no | 2,144 | 83.3 (81.3–85.2) | | 946 | 41.9 (39.7–44.2) | | 408 | 20.0 (18.2–22.0) | | 377 | 17.5 (15.4–20.0) | |
| yes | 480 | 83.2 (79.2–86.6) | | 259 | 50.4 (45.1–55.6) | | 69 | 13.0 (9.9–16.9) | | 89 | 15.0 (11.4–19.4) | |
| **Physical violence since the age of 16** | | | .746 | | | .971 | | | **.010** | | | **.003** |
| no | 2,311 | 84.0 (82.1–85.7) | | 1068 | 43.9 (41.8–46.0) | | 402 | 17.3 (15.6–19.1) | | 389 | 16.0 (14.0–18.2) | |
| yes | 223 | 83.0 (76.4–88.0) | | 98 | 43.8 (36.8–50.9) | | 51 | 25.6 (19.5–32.7) | | 61 | 25.8 (19.3–33.5) | |
| **Psychological violence since the age of 16** | | | .904 | | | **.019** | | | .386 | | | .448 |
| no | 1,997 | 83.9 (81.9–85.8) | | 893 | 42.5 (40.1–45.0) | | 349 | 17.6 (15.8–19.6) | | 341 | 16.6 (14.5–18.9) | |
| yes | 537 | 93.7 (79.8–86.9) | | 273 | 48.6 (44.3–52.9) | | 104 | 19.6 (16.0–23.7) | | 109 | 18.3 (14.4–23.0) | |
| **Physical violence by parent/ caregiver** | | | **.027** | | | **.001** | | | **.004** | | | **.016** |
| never | 1,638 | 82.0 (79.6–84.1) | | 812 | 46.6 (44.0–49.2) | | 266 | 16.2 (14.2–18.5) | | 263 | 15.1 (12.8–17.7) | |
| ever | 932 | 85.9 (83.3–88.2) | | 364 | 38.2 (34.4–42.1) | | 199 | 22.1 (19.1–25.3) | | 195 | 20.4 (17.0–24.3) | |
| **Psychological violence by parents/caregiver** | | | .256 | | | .482 | | | .212 | | | **.000** |
| never | 1,913 | 83.1 (81.1–85.0) | | 880 | 43.2 (40.8–45.7) | | 337 | 17.5 (15.6–19.7) | | 306 | 14.5 (12.4–16.9) | |
| ever | 631 | 85.4 (82.0–88.2) | | 287 | 45.2 (40.4–50.1) | | 124 | 20.2 (16.9–23.8) | | 144 | 23.0 (18.9–27.6) | |
| **Physical violence by peers** | | | .121 | | | .110 | | | .241 | | | .846 |
| never | 1,745 | 82.8 (80.7–84.7) | | 827 | 45.1 (42.6–47.7) | | 300 | 17.4 (15.4–19.6) | | 307 | 17.0 (14.8–19.4) | |
| ever | 795 | 85.5 (82.5–88.0) | | 344 | 41.3 (37.4–45.2) | | 154 | 19.9 (16.6–23.7) | | 139 | 16.6 (13.3–20.5) | |
| **Psychological violence by peers** | | | .522 | | | **.001** | | | .255 | | | .883 |
| never | 1,810 | 83.4 (81.2–85.3) | | 790 | 41.4 (38.7–44.0) | | 324 | 18.6 (16.7–20.8) | | 320 | 16.6 (14.4–19.1) | |

*(Continued)*

**Table 4.** (Continued)

| | Sexual and reproductive health outcome | | | | | | | | | | | |
| | Birth control pill (ever) | | | Current contraceptive use | | | Ever miscarriage or stillbirth | | | Ever abortion | | |
| Violence | n, yes | Weighted* % (95%- CI) | Weighted* $\chi^2$ | n, yes | Weighted* % (95% CI) | Weighted* $\chi^2$ | n, yes | Weighted* % (95% CI) | Weighted* $\chi^2$ | n, yes | Weighted* % (95%- CI) | Weighted* $\chi^2$ |
|---|---|---|---|---|---|---|---|---|---|---|---|---|
| ever | 727 | 84.6 (81.4–87.3) | | 384 | 50.3 (46.0–54.6) | | 128 | 16.5 (13.8–19.7) | | 121 | 16.9 (13.6–20.9) | |

\* Population weights according to [36]; N = absolute number of women who answered yes; 95%CI = 95% Confidence Interval; Significance is based on the adjusted F and its degrees of freedom. The adjusted F is a variant of the second-order Rao-Scott adjusted chi-square statistic.

psychological violence during childhood experiences perpetrated by peers and current contraceptive use (p>0.001) with women who responded to have been exposed to psychological peer violence indicating more often to currently use contraceptives (50.3%, 95%CI = 46.0–54.6%) than women who were never exposed to this type of violence (41.4% 95%CI = 38.7–44.0%).

Regarding miscarriage or stillbirth, a significantly lower prevalence (p>0.05) was found among women who reported to have been exposed to psychological violence in the previous 12 months of the survey (13.0%, 95%CI = 9.9–16.9%) than among women who indicated to not have experienced any incident of psychological violence in the past 12 months (20.0%, 95%CI = 18.2–22.0%). For women who responded to have experienced at least one incident of physical violence since the age of 16, a higher proportion for miscarriage or stillbirth was found (p>0.05; 25.6%, 95%CI = 19.5–32.7%) compared to women who did not experience any physical violence since they were 16 years old (17.3%, 95%CI = 15.6–19.1%). Additionally, 22.1% (95%CI = 19.1–25.3%) of women who reported physical violence by a parent or caregiver had at least one miscarriage or stillbirth, and this percentage was 16.2% (95%CI = 14.2–18.5%) among women who declared no physical violence by a parent/caregiver (p>0.05).

Women who declared to have been exposed to physical violence since the age of 16 had significantly more often (p>0.05) at least one abortion in their lives (25.8%, 95%CI = 19.3–33.5%) than those women who had not been exposed to physical violence since the age of 16 (16.0%, 95%CI = 14.0–18.2%). The proportion of women who ever had an abortion was also significantly higher (p>0.05) for women who reported any physical violence during childhood by a parent or caregiver with 20.4% (95%CI = 17.0–24.3%) compared to women who reported to never have experienced any physical violence by a parent or caregiver (15.1%, 95% CI = 12.8–17.7%). The prevalence of abortion differed significantly (p>0.001) between women who indicated to have experienced psychological violence during childhood by a parent/caregiver and those who did not.

## Multivariable analysis

Results of the crude logistic regression showed that women who reported to have experienced physical violence in the past 12 months were less likely to have ever taken the birth control pill with an odds ratio of 0.469 (95%CI = 0.252–0.875) compared to women who reported to not have been subject to physical violence in the 12 months prior to the survey, Table 5. After adjusting for age, marital status and SES (Model 1), the association did not remain statistically significant (aOR 0.587, 95%CI = 0.314–1.098). Social support and number of children (Model 2) as well as overall health status, alcohol consumption and chronic conditions (Model 3) did not seem to influence the association further. Hence, the fully adjusted model was not significant (aOR 0.618, 95%CI = 0.301–1.269). Moreover, compared to women who declared to never have experienced physical violence by a parent or another caregiver during childhood,

**Table 5. Models on sexual and reproductive health outcomes of violence.**

| | Sexual and reproductive health outcome | | | | | | | |
| --- | --- | --- | --- | --- | --- | --- | --- | --- |
| | Birth control pill (ever) | | Current contraceptive use | | Ever miscarriage or stillbirth | | Ever abortion | |
| **Violence** | Weighted* OR (95% CI) | Weighted* aOR (95% CI) | Weighted* OR (95% CI) | Weighted* aOR (95% CI) | Weighted* OR (95% CI) | Weighted* aOR (95% CI) | Weighted* OR (95% CI) | Weighted* aOR (95% CI) |
| **Physical violence 12 months** no | 1 | 1 | 1 | 1 | 1 | 1 | 1 | 1 |
| yes | **0.47 (0.25–0.88)** | 0.62 (0.30–1.27) | 1.38 (0.83–2.29) | 0.85 (0.42–1.69) | 0.70 (0.31–1.58) | 1.28 (0.50–3.31) | 0.93 (0.42–2.07) | 1.48 (0.58–3.77) |
| **Psychological violence 12 months** no | 1 | 1 | 1 | 1 | 1 | 1 | 1 | 1 |
| yes | 0.99 (0.73–1.35) | 1.00 (0.70–1.43) | **1.40 (1.11–1.78)** | 1.05 (0.77–1.44) | **0.60 (0.43–0.83)** | 0.80 (0.53–1.20) | 0.83 (0.59–1.16) | 1.03 (0.68–1.63) |
| **Physical violence since the age of 16** no | 1 | 1 | 1 | 1 | 1 | 1 | 1 | 1 |
| yes | 0.93 (0.60–1.45) | 0.87 (0.53–1.42) | 1.00 (0.74–1.34) | 0.86 (0.55–1.35) | **1.65 (1.13–2.40)** | **1.89 (1.17–3.04)** | **1.83 (1.23–2.71)** | 1.61 (0.95–2.74) |
| **Psychological violence since the age of 16** no | 1 | 1 | 1 | 1 | 1 | 1 | 1 | 1 |
| yes | 0.98 (0.72–1.33) | 0.93 (0.64–1.34) | **1.28 (1.04–1.57)** | 0.99 (0.76–1.30) | 1.13 (0.85–1.51) | **1.56 (1.06–2.30)** | 1.13 (0.83–1.54) | 1.45 (0.98–2.13) |
| **Physical violence by parent/caregiver** never | 1 | 1 | 1 | 1 | 1 | 1 | 1 | 1 |
| ever | **1.34 (1.03–1.74)** | **1.36 (1.02–1.81)** | **0.71 (0.58–0.87)** | 0.81 (0.62–1.04) | **1.46 (1.13–1.89)** | 1.20 (0.91–1.60) | **1.44 (1.07–1.94)** | 1.09 (0.79–1.50) |
| **Psychological violence by parents/caregiver** | | | | | | | | |
| never | 1 | 1 | 1 | 1 | 1 | 1 | 1 | 1 |
| ever | 1.18 (0.89–1.58) | 1.32 (0.94–1.85) | 1.08 (0.87–1.35) | 0.94 (0.73–1.20) | 1.19 (0.91–1.56) | **1.56 (1.06–2.30)** | **1.76 (1.29–2.39)** | **1.87 (1.30–2.70)** |
| **Physical violence by peers** never | 1 | 1 | 1 | 1 | 1 | 1 | 1 | 1 |
| ever | 1.22 (0.95–1.57) | 1.06 (0.79–1.42) | 0.85 (0.70–1.04) | **0.64 (0.51–0.82)** | 1.18 (0.89–1.57) | 1.17 (0.86–1.60) | 0.97 (0.73–1.30) | 0.91 (0.66–1.25) |
| **Psychological violence by peers** never | 1 | 1 | 1 | 1 | 1 | 1 | 1 | 1 |
| ever | 1.10 (0.83–1.45) | 1.00 (0.72–1.37) | **1.44 (1.16–1.79)** | 0.91 (0.70–1.20) | 0.86 (0.67–1.11) | 1.19 (0.87–1.62) | 1.02 (0.76–1.37) | 1.40 (0.97–2.01) |

\* Population weights according to [36]; 95%CI = 95% Confidence Interval; OR = Odds ratio; aOR = adjusted Odds Ratio, adjusted for age, marital status, socioeconomic status, social support, number of children, health status, alcohol consumption, chronic conditions; 1 = Reference group

those who reported to have experienced at least one incident of physical violence by a parent/caregiver were 1.3 times (OR 1.338, 95%CI = 1.033–1.735) more likely to have ever taken the birth control pill for contraception in the crude model. The association remained statistically significant, independently of age, marital status, SES, social support, number of children, health status, alcohol consumption and chronic conditions (aOR 1.356, 95%CI = 1.015–1.810).

When comparing women who indicated to have experienced some form of psychological violence in the past 12 months to women who answered to have not been exposed to any psychological violence in the previous 12 months in the crude model, the women who did experience psychological violence were 1.4 times (OR 1.404, 95%CI = 1.109–1.778) more likely to currently have used any method of contraception at the time of the survey. This significance disappeared after adjusting for age, marital status and SES (Model 1), and the fully adjusted

model was also not significant (aOR 1.051, 95%CI = 0.768–1.437). The crude results further suggested that women who reported to have been exposed to some form of psychological violence since the age of 16 had 1.2 times (OR 1.280, 95%CI = 1.041–1.574) higher odds for current contraceptive use than women who were not exposed. However, this relationship was no longer statistically significant in the fully adjusted model (aOR 0.992, 95%CI = 0.756–1.300).

Additionally, in the crude model, women who reported to have experienced psychological violence by peers during childhood had 1.4 times (OR 1.437, 95%CI = 1.155–1.786) higher chance for having answered to currently use contraceptives in the questionnaire than women who reported to never have been exposed to psychological peer violence. This association was not significant in the adjusted model for age, marital status and SES (aOR .913, 95%CI = .697–1.197).

Considering the associations between experiences of different forms of violence and miscarriage or stillbirth, results of the crude model showed that the chance for ever having a miscarriage or stillbirth was lower among women who declared to have experienced at least one incident which they classified as psychological violence in the past 12 months (OR 0.597, 95% CI = 0.430–0.829) than among women who declared that they did not. After the adjustment this association was not significant (aOR 0.800, 95%CI = 0.534–1.199). Furthermore, in the crude model, women who indicated to have been exposed to any physical violence since the age of 16 had a 1.6-fold (OR 1.645 (1.127–2.401) increased chances for having a miscarriage or stillbirth at some point in their lives compared to women who answered to never have been exposed to physical violence since they were 16 years old. This association stayed statistically significant in the fully adjusted model (aOR 1.887, 95%CI = 1.171–3.043).

Moreover, women that stated to never have been subject to physical violence by a parent or caregiver had an increased likelihood of ever having had a miscarriage or stillbirth (OR 1.463, 95%CI = 1.134–1.888) in the crude model, but this association was not significant in the fully adjusted model, namely after taking the confounders age, marital status and SES, into consideration (aOR 1.203, 95%CI = 0.905–1.600).

The crude results suggested that experiences of physical violence since the age of 16 were positively associated with the SRH outcome of ever having undergone an abortion. Women who reported to have experienced any physical violence in their lives had 1.8 times (OR 1.826, 95%CI = 1.230–2.711) higher chances for ever having had an abortion than women who did not. However, this relationship was not statistically significant after adjusting for age, marital status and SES in the full model (aOR 1.608, 95%CI = 0.945–2.735).

Additionally, women who stated to ever have been subject to physical childhood violence by a parent or caregiver were 1.4 times (OR 1.442, 95%CI = 1.071–1.941) more likely to ever have had an abortion in the crude model than women who stated to never have experienced any incidents of physical violence perpetrated by a parent or caregiver during childhood. This association also lost its significance after the adjustment for the socio-demographic factors, namely age, marital status and SES (aOR 1.086, 95%CI = 0.789–1.495).

Lastly, women who reported to have been exposed to psychological violence by a parent or caregiver were found to be almost 1.8 (OR 1.755, 95%CI = 1.291–2.385) times more likely to having at least one abortion at some point in their lives than women who did not report any psychological violence during childhood by a parent or caregiver. In the fully adjusted model this association remained significant (aOR 1.874, 95%CI = 1.299–1.702).

## Discussion

The objective of this work was to assess the magnitude of the association between different forms of violence experienced by women and sexual and reproductive health outcomes in Germany, using a nationally-representative sample.

We found that exposure to physical violence by a parent or caregiver increased the odds of lifetime birth control pill utilization, a miscarriage or stillbirth was more likely among women reporting exposure to physical violence since the age of 16, and an abortion was more likely among women reporting exposure to psychological violence by a parent or caregiver.

## Women experiencing violence

About 3.3% of women reported to have experienced physical violence and 20.2% have experienced psychological violence in the 12 months previous to the survey, confirming previous descriptions [33, 40].

The 12-month prevalence estimates for physical and psychological violence are lower than those found by other studies conducted among German women. Schröttle and colleagues found in their study for the German Federal Ministry for Family Affairs, Senior Citizens, women and Youth (BMFSFJ) (2004) [41], that about 7% of all surveyed women said they had experienced situations of physical violence in the last 12 months and 13.3% stated that they had experienced situations of psychological violence in the last 12 months. Also, the Fundamental Rights Agency (FRA) study (2014) [9] found that in Germany about 7% had experienced physical violence in the 12 months before the survey. The lower prevalence found might be explained by the differences in survey designs as FRA and BMFSFJ used a face-to-face interview in which participants were asked how often they have experienced specific situations of physical or psychological violence, such as pushing, slapping, insulting, yelling or humiliating. In DEGS1 participants were asked about having experienced any situation in which someone physically assaulted or psychologically hurt them in the past 12 months, without sub-questions, specific actions or acts described. This might have resulted in lower prevalence rates since participants might not have categorized some incidents as physical or psychological violence although they were or they did not report them as they were seen as not as severe enough to report them.

Regarding violence during childhood, the results are in line with national and international literature. The prevalence of physical and psychological violence by a parent or caregiver during childhood was, respectively, 36.0% and 25.9%, which is in line with the results from FRA (2014) [9], for Europe, where 35% of women experienced violence before the age of 15, and 27% having experienced physical and 10% having experienced psychological violence by an adult family member. For Germany, these figures were 44% of women who experienced some form of violence during childhood, including a prevalence of 37% for physical and 13% for psychological violence [9].

As illustrated in the results, approximately one third of women reported to ever have experienced physical and psychological violence during childhood perpetrated by peers, which could be considered an average value, when looking at the prevalence ranges of peer violence observed globally, that has been shown to vary between 15% to 50% [42].

Our results are in accordance with international research regarding factors that increase the probability of violence victimization for women. Violent incidents peak in women of reproductive age, with young women, under the age of 35 years, being particularly affected [8, 9, 12]. We also observed that 12-months physical and psychological violence was highest among women aged 18 to 24 and constantly decreased with higher age.

Also, our results point that women with a low SES were more likely to report violence than women with a high SES, which coincides with international evidence regarding the association between violence and economic, educational and occupational resources. The reasons for this relationship include a lack of financial independence which stuck women in abusive relationships, and that violence, in turn, might prevent women from seeking jobs [43]. Another

explanation is that violence and aggressions appear due to economic problems as low-income families experience higher tension and pressure than high-income families [44]. However, a high SES does not protect women from violence, which is supported by our results as violence occurred in all three categories of SES.

Alcohol consumption seemed also to increase the likelihood of women to report experiences of physical and psychological violence in the DEGS1 sample. Heavy drinking can be a source of conflict, encourage violent behavior and aggressiveness and alcohol consumption can be a coping mechanism for violence victimization [43, 45].

## Sexual and reproductive health

Women living in Germany, have a relatively good SRH compared to the global level, e.g., contraception is widely available, and few women face an unmet need for family planning [46]. As pointed out by BZgA (2018) [47] as well as Helfferich and colleagues (2016) [46] 67% to 78% of women used contraceptives in the 12 months prior to the surveys. These results coincide with the prevalence found for women who ever took the birth control pill, in the sample analysed (83.3%).

According to the official statistics by the Federal Statistical Office [48], about 100.000 women had an abortion in Germany in 2020. Helfferich and colleagues (2016) [46] moreover found that about 8.2% of women in their survey reported to ever have undergone an abortion in their lives. In the current study sample, more than twice as many women reported to have ever had an abortion (17.0%). A decrease in abortions has been described more recently in the country [48], which might explain the difference to the results presented, since the sample analysed was assessed earlier (between 2008 and 2011).

For miscarriage or stillbirth, a lifetime prevalence of 18.5% was found. Statistics on miscarriages are very limited in Germany because miscarriages are not subject to compulsory civil registration and therefore are not statistically documented. Data on miscarriages are difficult to collect as they mostly occur in the first few weeks of pregnancy. Thus, many women do not recognize that they were pregnant and interpret a miscarriage as an irregularity in their menstrual cycle. Nonetheless, the results coincide with the findings of Quenby and colleagues (2021) [49], who assumed that about 15.3% of all recognized pregnancies end in a miscarriage.

## Sexual and reproductive health outcomes associated with violence

While a range of studies found that women who experienced violence were less likely to use contraception [21, 22], other studies have reported contrary findings [26, 27] and some studies also did not find a significant relationship. Such mixed results appear possibly due to the timing in which violence occurred in relation to contraceptive use and various possible pathways linking the two [29]. These pathways include that women affected by violence might be more concerned about having children and use contraceptives to try to avoid pregnancy. On the other hand, women affected by violence might have less autonomy about their contraceptive use or may fear a violent reaction towards their contraceptive use [21, 29, 30, 50, 51].

According to the literature, exposure to violence increases the risk for miscarriage or stillbirth. Pallitto and colleagues (2013) [18] estimated from their findings that women who were affected by violence have an about 1.5 times higher risk for stillbirth and a 2 times higher risk for pregnancy loss than women who were never affected by violence. Results from German study populations indicate a correlation between pregnancy loss and a history of violence in adulthood [17] as well as between miscarriages and a history of violence in childhood [52]. Our results support these relations, by showing that women who reported to have experienced

violence since the age of 16 were also more likely to have a miscarriage or stillbirth than women who did not experience violence.

In our results, only women who answered to have experienced psychological violence during childhood by a parent or caregiver had higher odds of ever having had an abortion, after adjustment for potential confounders. Although non-significant, experiences of other forms of violence pointed in the same direction. This is in line with the results obtained by the WHO (2013) [2], showing that women who experienced violence were more than twice as likely to have an abortion than women who never experienced violence. In the study of Bitzker (2009) [53] using a German sample, violence against women was also significantly correlated with abortions and unwanted pregnancies. Möhler and colleagues (2008) [52] furthermore, found higher rates of abortions among German women who had traumatic experiences during childhood.

The frequency and severity of the adverse effects of violence experiences on SRH are higher in other world regions, namely Africa, Asia and Latina America, compared to Europe, where fewer studies exist [20]. This is a reflection of the higher prevalence of IPV observed (both lifetime and during the past-year) in these regions, as shown by the latest global estimates [54], of the different levels of access and adequacy of sexual and reproductive health services (particularly evident, for example, in the sub-Saharan region [55]), and also of different levels of women´s empowerment and access to education [56]. Future studies conducted in Europe should consider analyzing the relationships between violence experiences and SRH within women from specific regions or countries, which could potentially help disentangle relevant influences and design concrete recommendations.

## Strengths and limitations

The prevalence estimates obtained are difficult to compare to other existing estimates due to differences in survey instruments and methodology. Other studies often assessed other forms of violence, such as sexual violence or controlling behavior. The prevalence estimates are likely to underestimate violence as women are often reluctant to disclose.

The nationally-representative German health interview and examination survey for adults (DEGS1) resorted to validated and standardized screening instruments, such as version 2.0 of the SF-36 Health Survey or AUDIT C which ensures the quality of the data.

The screening instrument used for violence assessment was pretested and validated, although comparisons with other studies are limited For example, sexual violence and sexual harassment were not surveyed. This was justified because no adequate support structures were in place at the study site and hence participants should not be retraumatized through questions on sexual violence. Data on sexual violence could have given valuable insights, especially as it might have important direct and indirect consequences on SRH.

The questionnaire asked if women ever used birth control pill and current contraceptive, using two different questions, which we analyzed as separate characteristics. Therefore, we were not able to test the effect of violence experiences over the accumulation of contraceptive methods.

The violence module also did not include differentiated questions on degrees of severity of violent acts in contrast to survey instruments used in many national and international studies. Thus we could not analyze in which context or relationship dynamic violent acts were embedded e.g., whether violence was used as a systematic pattern of control or an exercise of power or whether it appeared in a situational escalation of violence [57]. This also does not allow to assess a potential dose-response effect of worse SRH consequences with increasing frequency and severity of violence.

Although violence experiences tend to be recurrent, we did not fit models adjusted for the concomitant presence of different violence types/periods, since our approach and interest aimed to disentangle the association between each violence type and the selected SRH outcomes.

Since our aim was to assess the association between ever experiencing different forms of violence and the lifetime experience of SRH outcomes, we did not restrict our analysis to the usual reproductive age period. Current contraceptive use could have been hypothesized to be present only during this period, however, we noted its use among women in the older age groups (S6 Table in S1 File) and adjusted all regression models for age to address such potential confounding. We cannot rule out the potential for residual confounding by unmeasured variables.

We also cannot rule out potential selection bias. People who were willing to disclose information about themselves and to participate in surveys, might differ from women that did not participate. Women who experienced violence might avoid questions on their experiences to prevent retraumatization and are therefore less likely to participate [58, 59]. For eligible women (aged 18–64), we compared age (5 groups) and socioeconomic status (3 groups–low, medium, high) of those with full valid answers and those with at least one missing in one of the violence questions analyzed (S12 Table in S1 File). The comparison suggests that full-respondents were slightly younger and more frequently from a high socioeconomic status, which might translate into an overall underestimation of violence (given previously known age and socioeconomic trends of violence victimization).

Measures taken to mitigate biases and limitations, included not explicitly mentioning the word violence in the questionnaire, questioning participants outside of their homes as well as other general quality and safety management measures such as adequate training of staff. Nevertheless, the actual violence prevalence rates are possibly much higher than those found in this study and the results found on the associations between violence victimization and SRH outcomes are likely to be underestimated.

Finally, the cross-sectional design of the study does not allow to draw conclusions on causality or effect directions as well as the temporal sequence of events for most of the associations.

These results have implications for public health research and policy. In Europe, and particularly in Germany, there may be less interest in the adverse impact that violence experiences have on SRH outcomes, because of the comparatively high level of development and access to sexual and reproductive specialized services. However, our results suggest that such harsh effects are noticeable when assessing women from the German general population, particularly when considering specific forms of violence experienced during women´s lifetime (e.g., physical and psychological violence since the age of 16). This also suggests the need to include violence assessment in clinical encounters, in particular for health services dedicated to women´s health.

## Conclusions and recommendations

There are important data gaps on violence in Germany as violence screening is not part of the regular health monitoring. Representative, disaggregated cross-sectional and longitudinal data needs to be collected frequently on the prevalence of violence experienced by women, including its various forms, frequency and severity of violent incidents as well as on its impacts on sexual and reproductive, mental and physical health in Germany. Such data is crucial for key actors to be able to develop and implement initiatives, policies and services that prevent and combat all forms of violence against women and girls. Violence against women as well as SRH

need to be recognized and treated as public health concerns and addressed through a multisectoral approach by researchers, policy makers and the healthcare sector.

## Supporting information

**S1 File S1 Table. DEGS1 translated questions on violence and sexual and reproductive outcomes (own translation). S2 Table.** Violence victimization in the past 12 months and socio-demographic characteristics. **S3 Table.** Violence victimization since the age of 16 and socio-demographic characteristics. **S4 Table.** Childhood violence by parent/caregiver and socio-demographic characteristics. **S5 Table.** Childhood violence by peers and socio-demographic characteristics. **S6 Table.** Contraceptive use and socio-demographic characteristics. **S7 Table.** Sexual and reproductive health and socio-demographic characteristics. **S8 Table.** Confounders of the association between violence and birth control pill (ever). **S9 Table.** Confounders of the association between violence and current contraceptive use. **S10 Table.** Confounders of the association between violence and miscarriage or stillbirth (ever). **S11 Table.** Confounders of the association between violence and abortion (ever). **S12 Table.** Comparison between participants with full valid information and those with at least one missing value in one of the violence variables analysed.
(DOCX)

## Author Contributions

**Conceptualization:** Antonia Marie Wellmann, Diogo Costa.

**Data curation:** Diogo Costa.

**Formal analysis:** Antonia Marie Wellmann, Diogo Costa.

**Supervision:** Diogo Costa.

**Writing – original draft:** Antonia Marie Wellmann.

**Writing – review & editing:** Diogo Costa.

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
