## [Decision Letter · Decision Letter 0]

9 Nov 2023

PONE-D-22-28534Violence against women in adulthood and childhood, and sexual and reproductive health outcomes: analysis of the German Health Interview and Examination Survey for Adults (DEGS1)PLOS ONE

Dear Dr. Costa,

Thank you for submitting your manuscript to PLOS ONE. After careful consideration, we feel that it has merit but does not fully meet PLOS ONE’s publication criteria as it currently stands. Therefore, we invite you to submit a revised version of the manuscript that addresses the points raised during the review process.

We look forward to receiving your revised manuscript.

Kind regards,

Yitagesu Habtu Aweke, Ph.D

Academic Editor

PLOS ONE

Journal Requirements:

Additional Editor Comments:

In addtion to the reviwer, Could you please include more elaboration for your 'multivariate" logistic regression ? Is that really a multi-variate analysis ?

Reviewers' comments:

Reviewer's Responses to Questions

**Comments to the Author**

1. Is the manuscript technically sound, and do the data support the conclusions?

Reviewer #1: Yes

2. Has the statistical analysis been performed appropriately and rigorously? 

Reviewer #1: No

3. Have the authors made all data underlying the findings in their manuscript fully available?

Reviewer #1: No

4. Is the manuscript presented in an intelligible fashion and written in standard English?

Reviewer #1: Yes

5. Review Comments to the Author

Reviewer #1: Title

Violence against women in adulthood and childhood, and sexual and reproductive health outcomes: analysis of the German Health Interview and Examination Survey for Adults (DEGS1)

Change the title to

The sexual and reproductive health outcomes of women who experienced violence in adulthood and childhood in Germany.

Abstract

• Remove all the words abbreviation in the study abstract except “aOR”

• Line 16 to 17 should be rewritten as “ this study examined the association between women exposure to physical violence, psychological violence and sexual & reproductive health outcomes………..

• Collapse the study design under methods. Then the method section should read as “This study used a cross-sectional research design to collect data on violence……….

• I am not too comfortable with the use of this repeated word “violence against” instead use the word “women who experienced violence….”

• The level of significant should be stated in the abstract

• Change “confidence interval” to CI

• Change “negative” to adverse.

Introduction

• Wrong journal reference style. Use PLoS recommended reference formatting.

• More information is needed in the introduction section on how IPV relates to other SRH variables included in this study. More emphasizes was made on contraceptives neglecting other variables. At the same times it’s important to give some fact on other regional prevalence and the need for this study as a global health issue

• Lastly in the last paragraph please tell us how the study will contribute to global public health needs in relation to SRH

Methods

I have so much reservations on the operationalization of the outcome variables

- This study is on SRH outcome and beware that IPV is also part of SRH dimensions (This means there could some elements of collinearity) – You must provide the collinearity test

- The SRH is measure between age 15-49 covering the fecundity period and also to ensure better analytical output of unskewed dataset – Dataset must be restricted to this age group

- Birth control pill is a form of modern contraceptive which could have been added to those using contraceptive – how are you able to unmerge these participants

- Include how to access the dataset and permission needed

Results

- There is a need to stick with weighted and delete all unweighted information

- Missing values must be dropped because there is no way this can be useful to your results or literature needs

- Can we have a graph displaying the weighted percentage of all the outcome variables?

Discussion

- There are a lot of acronyms here that has not been defined previously e.g AUDIT C or CASMIN

- Change goal to objective

- For quantitative study we don’t explore we examine, assess etc

- Write the summary of your result immediately after the objective

- Remove all the “ (95%CI= 2.6 – 4.2%, weighted estimates)” make your discussion using “likelihood” “higher odds” “less likely” etc

- Compare your study to studies conducted in Africa where these issues are arguably high

Other comments

• The strengths and limitation section is too long – concise it

• Remove references

• I didn’t see any conclusion and recommendation section

• After strengths and limitations section before the conclusions and recommendations section, write a section called “implication for public health research and policy” here discuss the implications of the findings from this study to public health research and policy

6. PLOS authors have the option to publish the peer review history of their article (what does this mean?). If published, this will include your full peer review and any attached files.

Reviewer #1: **Yes: **Obasanjo Bolarinwa

---

## [Author Response · Author response to Decision Letter 0]

19 Nov 2023

Dear Editor,

Thank you for the opportunity to revise our manuscript: “Violence against women in adulthood and childhood, and sexual and reproductive health outcomes: analysis of the German Health Interview and Examination Survey for Adults (DEGS1)”. 

Please find below a point-by-point response to the reviewers concerns and find enclosed the revised version of the manuscript, where amendments are highlighted using tracked changes. A clean version is also enclosed.

We believe the changes proposed by the reviewers significantly improved the quality of our manuscript and we thank them.

Data Availability statement:

The data that support the findings of this study are owned by a third-party organization. The data are available from the Robert Koch Institute (available as Scientific Use File) but restrictions apply to the availability of these data, which were used under license for the current study, and so are not publicly available. Further information including instructions on how to request a Scientific Use File of these data can be found in the RKI website (https://www.rki.de/DE/Content/Forsch/FDZ/Zugang/SUF.html).

We hope this version complies with the standards of PLOS ONE for publication.

Yours sincerely,

Diogo Costa

Department of Population Medicine and Health Services Research, 

School of Public Health, Bielefeld University, 

P.o. Box 10 01 31, D- 33501 Bielefeld, Germany

Additional Editor Comments:

In addition to the reviewer, Could you please include more elaboration for your 'multivariate" logistic regression ? Is that really a multi-variate analysis ?

Response: We thank the Editor for this remark, and we now refer to our models as multivariable. Our models are multivariable or multiple logistic regression models because each model has one outcome variable and multiple predictors. 

We changed “multivariate” for “multivariable” in the abstract (page 2, line 27 of the revised manuscript), methods (page 11, lines 251 and detailed in lines 260-261), and results sections (page 19, line 381).

Reviewers' comments:

Reviewer #1: 

1. Title

Change the title to

The sexual and reproductive health outcomes of women who experienced violence in adulthood and childhood in Germany.

Response: We thank the reviewer suggestion and followed the recommendation for both the full and short titles. We still would like to keep the statement regarding the analysis conducted in the scope of the German Health Interview and Examination Survey for Adults. We, therefore, changed the full title to:

“Sexual and reproductive health outcomes of women who experience violence in adulthood and childhood in Germany: analysis of the German Health Interview and Examination Survey for Adults (DEGS1)”, and the short title to: “Sexual and reproductive health outcomes of women who experienced violence in Germany”. 

2. Abstract

Remove all the words abbreviation in the study abstract except “aOR”

Response: We removed SRH and DEGS1 abbreviations from the abstract and kept “aOR”.

3. Line 16 to 17 should be rewritten as “ this study examined the association between women exposure to physical violence, psychological violence and sexual & reproductive health outcomes………..

Response: We rephrased line 17 as suggested – “this study examined the association between women exposure to physical violence, psychological violence and sexual and reproductive health outcomes”, (lines 19-21 of the revised manuscript).

4. Collapse the study design under methods. Then the method section should read as “This study used a cross-sectional research design to collect data on violence……….

Response: We inserted the study design under methods as suggested – “This study used a cross-sectional research design to collect data on violence…” (line 24).

5. I am not too comfortable with the use of this repeated word “violence against” instead use the word “women who experienced violence….”

Response: The term violence against women corresponds to the terminology used by the World Health Organisation, the United Nations, the Council of Europe, among several other organizations, and is embedded in international and regional legislation and relevant policy documents. We, therefore, changed the term as suggested in several places throughout the manuscript, but also kept it in parts of the text that refer to other studies or reports where it is used.

6. The level of significant should be stated in the abstract

Response: Significance level was added to the abstract (p<0.05), line 32.

7. Change “confidence interval” to CI

Response: Changed accordingly.

8. Change “negative” to adverse.

Response: Changed accordingly, line 39.

9. Introduction

Wrong journal reference style. Use PLoS recommended reference formatting.

Response: The manuscript is now formatted using PLOS style.

10. More information is needed in the introduction section on how IPV relates to other SRH variables included in this study. More emphasizes was made on contraceptives neglecting other variables. At the same times it’s important to give some fact on other regional prevalence and the need for this study as a global health issue

Response: Two paragraphs were added to the introduction section, emphasising the associations between IPV and other SRH outcomes, also considering this problem as a global issue, resorting to a recently published longitudinal study and a systematic review. The specific text and references added were (page 4, lines 74-85):

“In a longitudinal analysis conducted in Australia during 2021-2022 among two cohorts, IPV exposure was shown associated with increased odds of experiencing several sexual and reproductive health outcomes, including infertility, termination and miscarriage, which showed a dose-response effect, i.e., increased odds of experience adverse outcomes with greater exposure to IPV [1]. 

A recent systematic review on IPV and perinatal health included 15 studies conducted in Asia, 12 in North America and Oceania, 12 in Central and South America, 6 in Africa and 5 in Europe, to study the relationship between IPV and 39 different outcomes [2]. Amongst the studies classified with high methodological quality, ten studies included miscarriage as an outcome and six of them found an association with IPV (with miscarriage risks ranging between 1.7 and 5.7 times higher among women experiencing IPV). In the same review, among seven studies with high methodological quality that reported outcomes of perinatal death, six studies associated IPV experiences with this outcome (with risk ranging between 1.4 and 6.04 times higher)”.

References:

1. Hutchinson M, Cosh SM, East L. Reproductive and sexual health effects of intimate partner violence: A longitudinal and intergenerational analysis. Sex Reprod Healthc. 2023;35: 100816. doi:10.1016/j.srhc.2023.100816

2. Pastor‐Moreno G, Ruiz‐Pérez I, Henares‐Montiel J, Escribà‐Agüir V, Higueras‐Callejón C, Ricci‐Cabello I. Intimate partner violence and perinatal health: a systematic review. BJOG An Int J Obstet Gynaecol. 2020;127: 537–547. doi:10.1111/1471-0528.16084

11. Lastly in the last paragraph please tell us how the study will contribute to global public health needs in relation to SRH

Response: The following sentence was added to the last paragraph of the introduction, emphasizing the need to address IPV and its association with adverse SRH as a global public health issue, for which the current contextualized results can contribute (page 5, lines 104-107):

“These contextualized results can help raise awareness about the adverse effects that violence experiences have in SRH outcomes, and about the need to address this global public health issue through a multisectoral approach that aims at violence prevention”.

12. Methods

I have so much reservations on the operationalization of the outcome variables

This study is on SRH outcome and beware that IPV is also part of SRH dimensions (This means there could some elements of collinearity) – You must provide the collinearity test

Response: In line with the studies cited and the more recent references added, our objective is to measure the magnitude of the association between IPV and SRH and treating different forms of IPV as single determinants of separate adverse SRH outcomes. The fact that IPV may be part of a dimension of SRH shows its pervasiveness and the need to prevent it. 

Since we stratified our analysis according to each type of violence assessed and each SRH outcome, we did not assess multicollinearity among these variables - models were fitted separately for each SRH outcome and each type of violence assessed. Therefore, Table 5 in the manuscript shows the results of the 32 separate final adjusted models (8 violence types X 4 SRH outcomes). This clarification was added to the methods section, page 12, lines 279-285.

13. The SRH is measure between age 15-49 covering the fecundity period and also to ensure better analytical output of unskewed dataset – Dataset must be restricted to this age group

Response: We thank the reviewer observation and acknowledge the normal period when these outcomes are assessed. However, we treated violence experiences as the dependent variable in our study and analysed the lifetime report of each SRH. Our interest was to look into women who had ever experienced, for example a miscarriage or stillbirth or abortion, in their lifetime. Acknowledging the limitations of the cross-sectional analysis to infer about the direction of the associations, we investigated the distribution of each SRH outcome according to the demographics characteristics of participants and noted that all of them were observed across all the age groups analysed (18-64 years), as shown in Supplementary Tables S6 and S7. The only characteristic analysed for which the questioning referred to women’s current situation (i.e. at the time of interviewing), was “current contraceptive use”, which could be hypothesised to be present only during the fecundity period. However, we noted that current contraceptive use was reported by 255 women in the age group 45-54 years old and by 14 women in the age group 55-64 years old (Supplementary Table S6). We, therefore, included all women in the analysis, and adjusted our models for age to address the potential confounding from the age distribution of this characteristic. We added this explanation to the Strengths and limitations section of the manuscript (page 29, lines 599-603).

14. Birth control pill is a form of modern contraceptive which could have been added to those using contraceptive – how are you able to unmerge these participants

Response: We thank the reviewer observation and acknowledge this as a limitation in our study. The questionnaire asked if women ever used birth control pill and current contraceptive, using two different questions, which we analysed as separate characteristics. Therefore, we were not able to test the effect of violence experiences over the accumulation of contraceptive methods. This was added to the Strengths and limitations section of the manuscript (page 29, lines 582-584).

15. Include how to access the dataset and permission needed

Response: The following data availability statement was shared:

The data that support the findings of this study are owned by a third-party organization. The data are available from the Robert Koch Institute (available as Scientific Use File) but restrictions apply to the availability of these data, which were used under license for the current study, and so are not publicly available. Further information including instructions on how to request a Scientific Use File of these data can be found in the RKI website (https://www.rki.de/DE/Content/Forsch/FDZ/Zugang/SUF.html).

16. Results

There is a need to stick with weighted and delete all unweighted information

Response: Accordingly, we deleted the columns correspondent to unweighted frequencies from Tables 1, 2 and 3. 

17. Missing values must be dropped because there is no way this can be useful to your results or literature needs

Response: We removed lines with missing values from Tables 1, 2 and 3.

18. Can we have a graph displaying the weighted percentage of all the outcome variables?

Response: Two graphs were added to the results section of the manuscript:

Figure 1. Violence experiences (%, weighted), page 15.

Figure 2. Sexual and reproductive health outcomes (%, weighted), page 16.

19. Discussion

- There are a lot of acronyms here that has not been defined previously e.g AUDIT C or CASMIN

Response: We checked all acronyms and added full names where missing. AUDIT C - Alcohol Use Disorders Identification Test-Consumption, was presented in the methods sections (page 9, lines 187-188). CASMIN was removed since it was not relevant in the analysis. We thank the reviewer for these corrections. 

20. Change goal to objective

Response: Changed accordingly (page 24, line 452). 

21. For quantitative study we don’t explore we examine, assess etc

Response: We changed “explore” to “assess”, page 24, line 452.

22. Write the summary of your result immediately after the objective

Response: a short summary of the main results was added after the objectives, page 24, lines 455-458. The text added was as follows:

“We found that exposure to physical violence by a parent or caregiver increased the odds of lifetime birth control pill utilization, a miscarriage or stillbirth was more likely among women reporting exposure to physical violence since the age of 16, and an abortion was more likely among women reporting exposure to psychological violence by a parent or caregiver.” 

23. Remove all the “ (95%CI= 2.6 – 4.2%, weighted estimates)” make your discussion using “likelihood” “higher odds” “less likely” etc

Response: We removed mentions to confidence intervals of estimates and used qualifiers as suggested throughout the discussion section.

24. Compare your study to studies conducted in Africa where these issues are arguably high

Response: The following paragraph and references were added to the discussion section to highlight this point (page 28, lines 559-565):

“The frequency and severity of the adverse effects of violence experiences on SRH are higher in other world regions, namely Africa, Asia and Latina America, compared to Europe, where fewer studies exist [1]. This is a reflection of the higher prevalence of IPV observed (both lifetime and during the past-year) in these regions, as shown by the latest global estimates [54], of the different levels of access and adequacy of sexual and reproductive health services (particularly evident, for example, in the sub-Saharan region [2]), and also of different levels of women´s empowerment and access to education [3].”

References:

1. Pastor‐Moreno G, Ruiz‐Pérez I, Henares‐Montiel J, Escribà‐Agüir V, Higueras‐Callejón C, Ricci‐Cabello I. Intimate partner violence and perinatal health: a systematic review. BJOG An Int J Obstet Gynaecol. 2020;127: 537–547. doi:10.1111/1471-0528.16084

2. Rosário EVN, Gomes MC, Brito M, Costa D. Determinants of maternal health care and birth outcome in the Dande Health and Demographic Surveillance System area, Angola. PLoS One. 2019;14. doi:10.1371/journal.pone.0221280

3. Garcia-Moreno C, Amin A. Violence against women: where are we 25 years after ICPD and where do we need to go? Sex Reprod Heal matters. 2019;27: 1–3. doi:10.1080/26410397.2019.1676533

25. Other comments

The strengths and limitation section is too long – concise it

Response: The strengths and limitations section was shortened following the reviewer suggestion. 

26. Remove references

Response: We would prefer not to remove the references used, as this is not a restriction listed in the journal's requirements.

27. I didn’t see any conclusion and recommendation section

Response: We added this title to the last section, that was meant to conclude and recommend on further actions. 

28. After strengths and limitations section before the conclusions and recommendations section, write a section called “implication for public health research and policy” here discuss the implications of the findings from this study to public health research and policy

Response: The following paragraph was added after the Strengths and limitations section and be

---

## [Decision Letter · Decision Letter 1]

27 May 2024

PONE-D-22-28534R1Sexual and reproductive health outcomes of women who experienced violence in adulthood and childhood in Germany: analysis of the German Health Interview and Examination Survey for Adults (DEGS1)PLOS ONE

Dear Dr. Costa,

Thank you for submitting your manuscript to PLOS ONE. After careful consideration, we feel that it has merit but does not fully meet PLOS ONE’s publication criteria as it currently stands. Therefore, we invite you to submit a revised version of the manuscript that addresses the points raised during the review process.

**ACADEMIC EDITOR: **Thank you for submitting your manuscript. Please find attached comments from the reviewers. It is important that you address all comments provided by the reviewers. In my reading, there are clearly other studies/manuscripts that have been written using the same data sources you have used but you have not considered these in your manuscript. These manuscripts will be helpful in contextualising your results and situating them along what others have found. See for example; Hunzelar C, Krumpholtz Y, Schlack R, Weltermann B. More GP Consultations by Violence Victims: Results from the Representative German DEGS1 Study. Int J Environ Res Public Health. 2023 Mar 6;20(5):4646. doi: 10.3390/ijerph20054646. PMID: 36901654; PMCID: PMC10001473.

We look forward to receiving your revised manuscript.

Kind regards,

Hanani Tabana, Ph.D

Academic Editor

PLOS ONE

Journal Requirements:

Additional Editor Comments:

Dear Diogo

Thank you for submitting your manuscript. Please find attached comments from the reviewers. It is important that you address all comments provided by the reviewers. In my reading, there are clearly other studies/manuscripts that have been written using the same data sources you have used but you have not considered these in your manuscript. These manuscripts will be helpful in contextualising your results and situating them along what others have found. See for example; Hunzelar C, Krumpholtz Y, Schlack R, Weltermann B. More GP Consultations by Violence Victims: Results from the Representative German DEGS1 Study. Int J Environ Res Public Health. 2023 Mar 6;20(5):4646. doi: 10.3390/ijerph20054646. PMID: 36901654; PMCID: PMC10001473.

Looking forward to receiving your revised manuscript.

Best wishes, Hanani

Reviewers' comments:

Reviewer's Responses to Questions

**Comments to the Author**

1. If the authors have adequately addressed your comments raised in a previous round of review and you feel that this manuscript is now acceptable for publication, you may indicate that here to bypass the “Comments to the Author” section, enter your conflict of interest statement in the “Confidential to Editor” section, and submit your "Accept" recommendation.

Reviewer #2: (No Response)

Reviewer #3: All comments have been addressed

2. Is the manuscript technically sound, and do the data support the conclusions?

Reviewer #2: (No Response)

Reviewer #3: Yes

3. Has the statistical analysis been performed appropriately and rigorously? 

Reviewer #2: (No Response)

Reviewer #3: Yes

4. Have the authors made all data underlying the findings in their manuscript fully available?

Reviewer #2: (No Response)

Reviewer #3: Yes

5. Is the manuscript presented in an intelligible fashion and written in standard English?

Reviewer #2: (No Response)

Reviewer #3: Yes

6. Review Comments to the Author

Reviewer #2: (No Response)

Reviewer #3: I like the author’s integration of Sexual and reproductive health outcomes and violence. I am not sure whether it's necessary to include “adulthood and childhood” in the title. The author could leave the two concepts out of the title and highlight them in the work, explaining them further in the methodology. Explain a bit more about what you mean by multivariate logistic regression.

7. PLOS authors have the option to publish the peer review history of their article (what does this mean?). If published, this will include your full peer review and any attached files.

Reviewer #2: **Yes: **ANGEL PHUTI

Reviewer #3: **Yes: **JOYCE OMWOHA

---

## [Author Response · Author response to Decision Letter 1]

3 Jun 2024

Academic Editor

In my reading, there are clearly other studies/manuscripts that have been written using the same data sources you have used but you have not considered these in your manuscript. These manuscripts will be helpful in contextualising your results and situating them along what others have found. See for example; Hunzelar C, Krumpholtz Y, Schlack R, Weltermann B. More GP Consultations by Violence Victims: Results from the Representative German DEGS1 Study. Int J Environ Res Public Health. 2023 Mar 6;20(5):4646. doi: 10.3390/ijerph20054646. PMID: 36901654; PMCID: PMC10001473.

Response: We have added this study as a reference (ref #31) and a brief sentence about its main results to the introduction section of our revised manuscript (page 6, lines 103-109). Please note that we submitted our manuscript to PLOS ONE in October 2022, when this article had not yet been published.

Review Comments to the Author

I like the author’s integration of Sexual and reproductive health outcomes and violence. I am not sure whether it's necessary to include “adulthood and childhood” in the title. The author could leave the two concepts out of the title and highlight them in the work, explaining them further in the methodology. Explain a bit more about what you mean by multivariate logistic regression.

Response: We thank the reviewer comment and removed the terms “adulthood and childhood” from the title, while clarifying how these were defined in the methods section (page 9, line 177).

In the previous round of reviews, we corrected the term multivariate for multivariable, as this was what we did: our models are multivariable or multiple logistic regression models because each model has one outcome variable and multiple predictors. This was added to the methods section (page 12, lines 269-270).

AP Review

Abstract

1. Line 21: and SRH outcomes- SRH appears for the first time as an abbreviation without it written in full anywhere else (above).

Response: We thank the reviewer for the careful reading of our manuscript and suggestions. We revised accordingly, page 2, line 23 – Sexual and reproductive health (SRH).

2. Line 23: regression models were fitted to: how about ‘used’- fitting could be more relevant when describing data depicted in a plot.

Response: Changed accordingly throughout the manuscript (page 2 line 25, page 12 line 260, and page 13 line 280).

3. This sentence from your conclusion ‘Exposure to violence was associated with adverse sexual and reproductive health outcomes among adult German women’, supported by the stats would make your results more tangiable/fully understood or ‘have more weight’. Otherwise it’s just a ‘too technical presentation’.

Response: We thank the reviewer suggestion and changed the sentence as follows (page 2, lines 35-37): “The results suggest that adult German women who experienced physical or psychological violence since the age of 16, including violence perpetrated by a parent or caregiver, were more likely to report miscarriage or stillbirth and abortion”.

4. Conclusion: 

The authors wrote: “Scoping for violence experiences in clinical encounters should be considered for the prevention of women’s adverse health outcomes.”

Scoping means ‘act or practice of eyeing or examining’ -According to websearch for example:

Considered by who- make it clear?

Clinical encounters are generally by the health care provider – so they don’t do it?

Response: We rephrased the sentence in the abstract to make it clearer and to answer the specific questions raised by the reviewer. The sentence now reads as follows (page 2, lines 37-43): “Direct assessment of violence experiences against women should be conducted by healthcare professionals in clinical encounters, particularly by obstetrics and gynaecological specialists, for the prevention of women´s adverse sexual and reproductive health outcomes. Furthermore, violence should be treated as a major public health concern and addressed through a multisectoral approach, involving the healthcare and educational sectors, researchers and relevant policymakers”. 

5. Hunzelar C, Krumpholtz Y, Schlack R, Weltermann B. More GP Consultations by Violence Victims: Results from the Representative German DEGS1 Study. Int J Environ Res Public Health. 2023 Mar 6;20(5):4646. doi: 10.3390/ijerph20054646. PMID: 36901654; PMCID: PMC10001473.

This study above on the DEGS1 Germany -2008-2011 reports on those experiencing violence n=3149 (females) … they reported that those who are victims of violence consulted quite often…but in your conclusion you indicate that the health practitioners should consider it. As a reader, this is the main conclusion drawn from your study and its somehow unclear.

This is the main conclusion of the study in discussion

The high frequency of GP contacts in VE victims constitutes opportunities to professionally support this vulnerable patient group and underlines the necessity for GPs to integrate VE as a bio-psycho-social problem in a holistic treatment approach.

Since they analysed n=3149 (females)- DEGS1 Germany -2008-2011 (Just like the authors) & from my understanding came to the same conclusion as in this manuscript regarding ‘clinical encounters’ How is this study different from the one already published?

Its clear you wish to highlight more on SRH make it more visible in your conclusions!

Where in the text is the obstetrics and gynaecologist for example, organisations dealing with violence… 

Response: We thank the reviewer’s remark and made changes accordingly. Specifically, we introduced the main result of the study by Hunzelar et al., (2023) in the introduction section of our revised manuscript (page 6 lines 102-105), to state that despite their findings, the study did not explore the relationship between women who experienced violence and sexual and reproductive health outcomes. We added a reference to the study. 

We also changed our conclusions in the abstract to more clearly reflect our findings, as suggested (page 2, lines 35-43): “The results suggest that adult German women who experienced physical or psychological violence since the age of 16, including violence perpetrated by a parent or caregiver, were more likely to report miscarriage or stillbirth and abortion. Direct assessment of violence experiences against women should be conducted by healthcare professionals in clinical encounters, particularly by obstetrics and gynaecological specialists, for the prevention of women´s adverse sexual and reproductive health outcomes. Furthermore, violence should be treated as a major public health concern and addressed through a multisectoral approach, involving the healthcare and educational sectors, researchers and relevant policymakers.”

6. Hunzelar C, Krumpholtz Y, Schlack R, Weltermann B. More GP Consultations by Violence Victims: Results from the Representative German DEGS1 Study. Int J Environ Res Public Health. 2023 Mar 6;20(5):4646. doi: 10.3390/ijerph20054646. PMID: 36901654; PMCID: PMC10001473.

Interestingly, the study above made an analysis of the same database and nowhere does it appear on this manuscript. 

Response: Indeed. This is because the cited study was not published when we submitted our manuscript to PLOS ONE. We have now integrated its findings in our introduction section (page 6, lines 102-105) and added the reference (ref #31). 

7. The conclusion part could target the results better especially that they ‘associations statistically significant’ where, the results variables /keywords and: Age of victim (young therefore still attending school most likely-) 

Parent 

caregivers

You want your research to make your research have a positive impact on health, right? Why put a recommendation towards a health care provider only? Advocate more on awareness of violence in e.g schools, now that you found out on the ‘parents’ contribution to the SRH adverse effects what do you recommend?

Response: We have now rephrased the conclusions to reflect the importance of these topics. Specifically, we added that our results suggest that “adult German women who experienced physical or psychological violence since the age of 16, including violence perpetrated by a parent or caregiver, were more likely to report miscarriage or stillbirth and abortion” (page 2, lines 35-37) and that “violence should be treated as a major public health concern and addressed through a multisectoral approach, involving the healthcare and educational sectors, researchers and relevant policymakers” (page 2-3, lines 41-43).

8. This is part of your manuscript conclusion:

Violence against women as well as SRH need to be recognized and treated as public health concerns and addressed through a multisectoral approach by researchers, policy makers and the healthcare sector…

consider such statements for your abstract conclusion

Response: Thank you. We have added such elements as mentioned above (pages 2-3, lines 35-43).

9. Methodology 

Line 114: such as residential homes [33,34] (n=8,152).

Consider position of ((n=8,152) … ? before references

Response: Relocated to line 119, thank you for spotting this.

10. Line 124….4,193: Rules on starting sentences with numbers (remember them)

Response: Changed the start of the sentence (line 131: “Of these, 4.193 were first-time participants”).

11. Results:

‘German citizens, people of foreign nationality and people living in institutions

such as residential homes’

The researcher anticipates a difference in outcomes between these population groups., influenced by SES, Origin, migration status among foreign national etc…

Any rationale for not doing a subgroup analysis or test of association between the variables above with SRH outcomes.

Response: The mentioned categories were used as inclusion criteria for participants when implementing the DEGS survey and were not assessed specifically for their potential relationship with different outcomes. Still, we included in our analysis several potential confounding factors (e.g. socioeconomic status, marital status, social support). However, we cannot rule out the potential for residual confounding by unmeasured variables, which is common to all observational studies. We added this limitation to our discussion section (page 30, lines 603-604).

12. Line 547 – 553 of your discussion

The frequency and severity of the adverse effects of violence experiences on SRH are higher in other world regions, namely Africa, Asia and Latina America, compared to Europe, where fewer studies exist [20]….

Don’t you think it worth to check outcomes of the population from these regions now living in Europe in your study?

Which nation experienced the worst outcomes? Or did the reviewer miss it, its somewhere? / rationale for not doing the analysis.

Ps: This helps in designing the appropriate / relevant Interventions/ recommendations to combat all negative outcome.

Response: The reviewer raises a very pertinent point. However, we did not access or analyzed variables related to participants country of origin. Still, we think this should be done in the future and added such statement to our discussion section (page 29, lines 569-571): “Future studies conducted in Europe should consider analyzing the relationships between violence experiences and SRH within women from specific regions or countries, which could potentially help disentangle relevant influences and design concrete recommendations”.

13. Discussion:

Line 462: survey designs as FRA and BMFSFJ: check that the 2 abbreviations were written in full somewhere before discussion.

Response: Added the full names accordingly, page 25, lines 473 and 476.

14. Line 576: two full stops-typo?

Response: Corrected, thank you.

15. Reference:

Numerous mistakes that are scientifically not OK! Please check every reference

Examples: 

Reference 1-9 , 31, 38, 39…

Response: Thank you for the careful reading. We have now revised all references.

---

## [Editor Report · Decision Letter 2]

10 Jun 2024

Sexual and reproductive health outcomes of women who experienced violence in Germany: analysis of the German Health Interview and Examination Survey for Adults (DEGS1)

PONE-D-22-28534R2

Dear Dr. Costa,

We’re pleased to inform you that your manuscript has been judged scientifically suitable for publication and will be formally accepted for publication once it meets all outstanding technical requirements.

Kind regards,

Hanani Tabana, Ph.D

Academic Editor

PLOS ONE
---

## [Editor Report · Acceptance letter]

2 Jul 2024

PONE-D-22-28534R2 

PLOS ONE

Dear Dr. Costa, 

I'm pleased to inform you that your manuscript has been deemed suitable for publication in PLOS ONE. Congratulations! Your manuscript is now being handed over to our production team.

Kind regards, 

on behalf of

Associate Professor Hanani Tabana 

Academic Editor

PLOS ONE